

# Could an event of extreme drought (2019-2020) affect the feeding ecology of *Bubo magellanicus* (Gmelin 1788) (Strigiformes: Strigidae) in a Mediterranean region of Chile?

Sam Catchpole Ahumada[1], Luis Carrera Suárez[2] and Reinaldo Rivera[3,4]

[1] Catchpole & Asociados, consultoría e investigación, Concepción, Biobío, Chile
[2] Caminando Otro Sendero E.I.R.L, Talcahuano, Biobío, Chile
[3] Instituto Milenio de Oceanografía, Universidad de Concepción, Concepción, Biobío, Chile
[4] Facultad de Ciencias Naturales y Oceanográficas, Laboratorio de Ecología Evolutiva y Filoinformática, Concepción, Biobío, Chile

## ABSTRACT

Global warming generates changes in environmental conditions, affecting the spatial-temporal dynamics of precipitation and temperature. Droughts, events of low rainfall, are becoming more frequent and severe. In central Chile, from 2010 to date, an unprecedented drought event has developed, affecting the ecosystem and creating pressure on the dynamics of food webs. The present study analysed the trophic ecology of *Bubo magellanicus*, a top predator in the Mediterranean region of Chile, between 2019 and 2020 a period with a rainfall deficit of 72.6%. Our results established a diet mainly described by invertebrates (97.75%), in particular by the *Gramnostola rosea* spider (87.86%), and a low contribution of small vertebrates (2.24%). The trophic niche breadth (B = 0.37) and the standardised Levin's index ($B_{STA}$ = 0.01) are the lowest recorded in the species *B. magellanicus*. A comparative analysis of trophic ecology with other studies developed in the same region established significant differences in the composition of the diet (frequency of occurrence of prey unit). This work provides evidence that droughts and other extreme environmental scenarios restructure the food webs of an ecosystem, with direct consequences on the trophic niche of the species, specifically top predators.

# INTRODUCTION

Droughts are natural hazards that result from a precipitation deficit with respect to what is considered "normal" (*Wilhite & Buchanan-Smith, 2005*; *Kiem et al., 2016*). According to the predictions of global climate change, there will be an increase in frequency, severity and duration of this natural hazards (*Dai, 2013*; *Satoh et al., 2022*), with more serious damages for anthropogenic activities and severe alteration in the environment (Wilhite & Buchanan-Smith 2005), with the latter probably being more important

Corresponding author
Reinaldo Rivera, reijavier@gmail.com

than the socioeconomic impacts (*van Dijk et al., 2013*; *Felbermayr et al., 2022*). However, determining the impact of a drought in environmental and ecological terms is complex, as affected species can be influenced by multiple ecological pathways including their interactions, such as competition, growth or survival rates (*Chesson & Huntly, 1997*); and alterations or absences of trophic levels of food webs (*Prugh et al., 2018*; *De Necker et al., 2022*). Other studies have shown that droughts strongly alter the dynamics of animal populations (*Cruz-McDonnell & Wolf, 2016*; *Lister & García, 2018*), affecting the survival and reproductive success of a wide variety of organisms, including mammals (*Bourne et al., 2020a*), reptiles (*Sperry & Weatherhead, 2008*; *Martín et al., 2022*) and birds (*Bourne et al., 2020b*), with important consequences in the community structure (*Ledger et al., 2013*) and severe effects on top predators due to the shift and removal of hug species (*Prugh et al., 2018*), with alterations in their diet and trophic niche (*Henschel & Skinner, 1990*; *Van Horne, Schooley & Sharpe, 1998*; *McDowell & Medlin, 2009*). This behaviour refers to the bottom-up control of community ecology, predicting that a decrease in the lower levels of the food web would affect predator populations negatively (*Hunter & Price, 1992*; *Sperry & Weatherhead, 2008*).

In top predators as raptors, their pellets have been used since the early 20th century as an important tool to investigate aspects of their diet, ecology and prey diversity (*e.g.*, *Fisher, 1896*; *Errington, 1930*). More recently, they have been used as indicators of prey abundance or surrounding animals (*McDowell & Medlin, 2009*; *Andrade, De Menezes & Monjeau, 2016*). However, scarce information has been obtained regarding the cause–effect relationship between the increase of drought due to climate change and its effects on the food ecology of raptors, especially owls, that are considered a potential estimator of abundance of surrounding animals (small mammals, birds and arthropods) and characterise middle levels assemblages of food webs (*McDowell & Medlin, 2009*; *Andrade, De Menezes & Monjeau, 2016*). *Bubo magellanicus* has a wide distribution in South America, in Chile ranges from 18°–54° S (*Jaksic & Marti, 1984*), is an eurytopic species, inhabiting forests, plantations, shrubs, deserts, mountain areas and coastal islands (*Humphrey et al., 1970*; *Burton, 1973*; *Mella et al., 2016*; *Udrizar Sauthier et al., 2017a*; *Udrizar Sauthier et al., 2017b*), including urban parks (*Jaksic et al., 2001*). Therefore, the heterogeneity of habitats where it lives presents a wide spectrum for prey selection, including small mammals (rodents and marsupials), birds, reptiles, insects and arachnids (*Jaksic & Marti, 1984*; *Jaksic, Yáñez & Rau, 1986*; *Trejo, Guthmann & Lozada, 2005*; *Nabte, Saba & Pardiñas, 2006*; *Ortiz et al., 2010*; *Muñoz Pedreros et al., 2017*; *Udrizar Sauthier et al., 2017a*; *Udrizar Sauthier et al., 2017b*; *Martínez, 2018*; *Vega, Jara & Mella, 2018*; *Cheli et al., 2019*; *Rau & Mansilla, 2019*), mainly preying on the most abundant species of the place (*Mella, 2002*; *Mella et al., 2016*). Nevertheless, there are some hypotheses suggesting that the consumption of some prey does not depend on their abundance, but on the energy cost that they may contribute to. Therefore, species weighing less than 20 g are seldomly preyed on and constitute a low percentage in the owl's diet (*Jaksic & Marti, 1984*) or in the predator–prey relation (*Trejo, Guthmann & Lozada, 2005*). In another scenario, such as in island ecosystems, with some isolation from the continental land mass, limited resources (*Udrizar Sauthier et al., 2017a*; *Udrizar Sauthier et al., 2017b*) and under extreme environmental conditions (*i.e.,* drought)

the feeding ecology of owls could differ from normal conditions in terms of prey selectivity and diversity, as well as diet composition, due to limited presence or replacement of their usual prey. The aim of this study was to describe the effects of a drought, as an extreme environmental scenario, on the feeding ecology of *B. magellanicus*, evaluating (i) prey selection, richness and diversity of prey units (PU) and surrounding species during the drought conditions, (ii) trophic niche breadth using diet descriptors and (iii) comparing other diet studies of the species in the Mediterranean region of Chile.

## MATERIAL AND METHODS

### Study area

The study was conducted in Las Tórtolas (33°9′41.50″S, 70°42′23.25″W), located in the Metropolitan Region of Chile (Fig. 1). The dominant vegetation of the area corresponds to Mediterranean scrub, specifically an Andean Mediterranean thorny forest of *Acacia caven* and *Baccharis paniculata* (*Luebert & Pliscoff, 2017*). During the study, the area was mostly dry with scarce vegetation associated and few isolated trees of the species *Prosopis chilensis* (Molina) Stunz emend Burkart. The climate of the study area is warm temperate, characterised by short periods of rain during winter (July–September) and a six to eight months (October–May) dry season (*sensu Köppen, 1948*). But, since 2010 to date, central Chile has been affected by an uninterrupted sequence of dry years (12 years), with annual rainfall deficits from 25 to 45% (*Garreaud et al., 2017*). This so-called Mega-Drought (MD) is the longest event on record in the last millennia in this Mediterranean-like region (*CR2, 2015*; *Garreaud et al., 2020*). The MD has co-occurred with the warmest decade on record for central Chile (*Vuille et al., 2015*), together with other factors increasing its severity, such as: the decrease of the Andean snowpack that affect rivers flow, diminution of reservoir volumes and groundwaters levels; the increase of evaporation reservoirs (*Dai, 2013*; *Garreaud et al., 2017*); and the reduction of vegetation productivity (*Zambrano et al., 2020*). During the study, the annual rainfall deficit reached 72.6% on average, with monthly deficits over 80% for January (100%), May (81.1%), July (82.2%), August (90.5%) and December (87.5%) (Fig. 2).

### Pellet collection and analysis

We collected and analysed the composition of pellets of the Magellanic Horned Owl, *Bubo magellanicus* (Gmelin 1788), in the Mediterranean region of Chile, which is considered a hotspot of biodiversity (*Myers et al., 2000*) that has been suffering a prolonged drought period for more than a decade (*Núñez et al., 2011*; *Boisier et al., 2016*; *CR2, 2015*; *Garreaud et al., 2017*; *Garreaud et al., 2020*), reaching critical levels of rainfall deficit between the years 2018 and 2019 (*Zambrano et al., 2020*).

Between April 2019 and January 2020, pellets from *B. magellanicus* were collected at the piedmont, using a carob tree as a perch. The pellets were removed in their entirety systematically every three months, sampling the different seasons of the year. The collected pellets were deposited in hermetic bags and labelled with the date and time for further analysis in the laboratory. Only the entire pellets were weighed and measured using a calliper (0.1 mm) and a digital weight (0.1 g) precision.

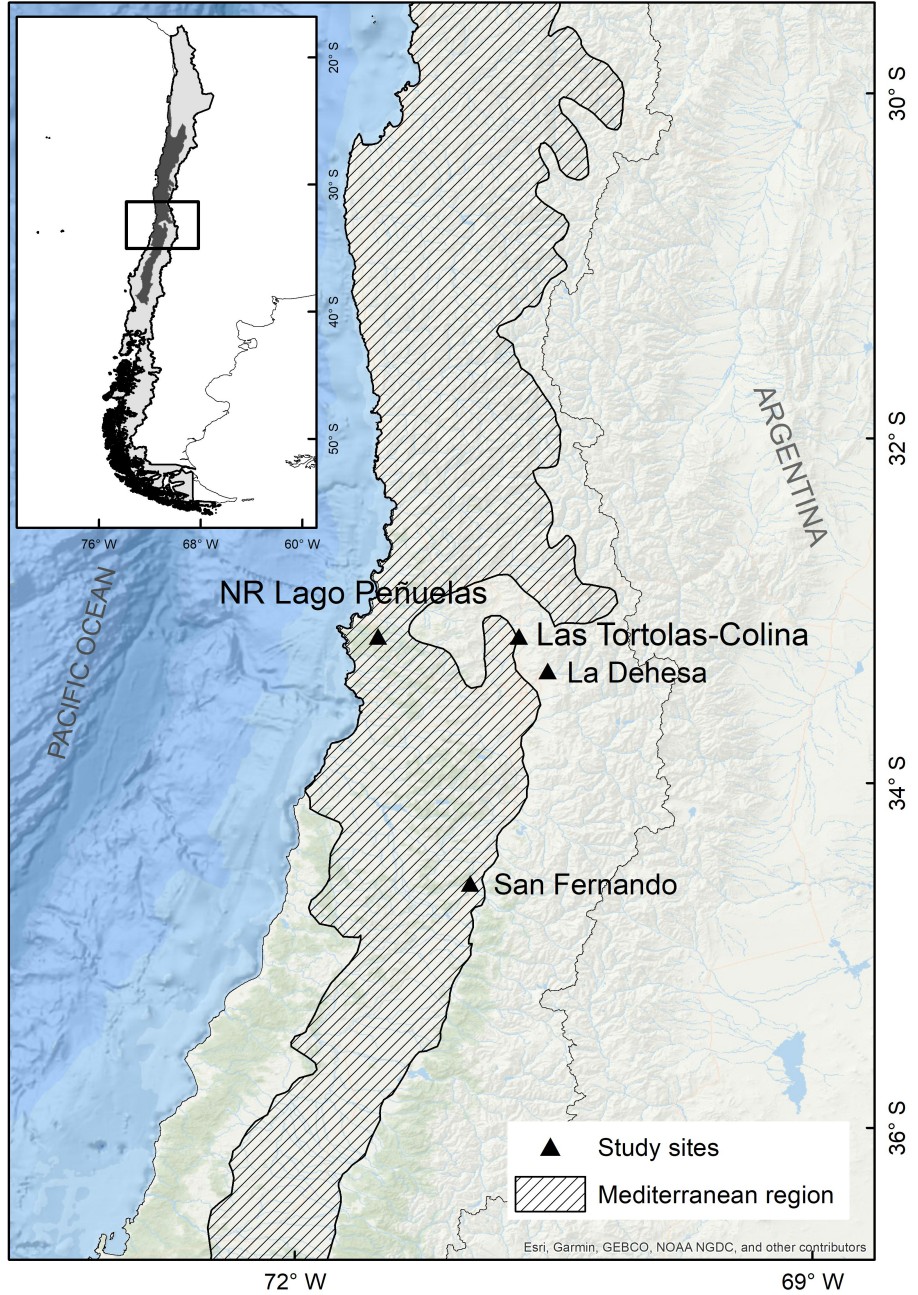

**Figure 1** Map of the mediterranean region in Chile and the collection sites of pellets of the present study (**Las Tórtolas**) and the rest of studies developed by *Yáñez, Rau & Jaksic (1978)* (**San Fernando**), *Jaksic & Yáñez, 1980* (**La Dehesa**) y *Muñoz Pedreros et al., 2017* (**Lago Peñuelas National Reserve**).

Pellets were manually disaggregated using warm water to dissolve the outer mucous coat, and all the remains of invertebrate, hairs, feathers, bones and other organic fragments were separated (*Marti, 1987*; *McDowell & Medlin, 2009*). For the identification, we used keys and

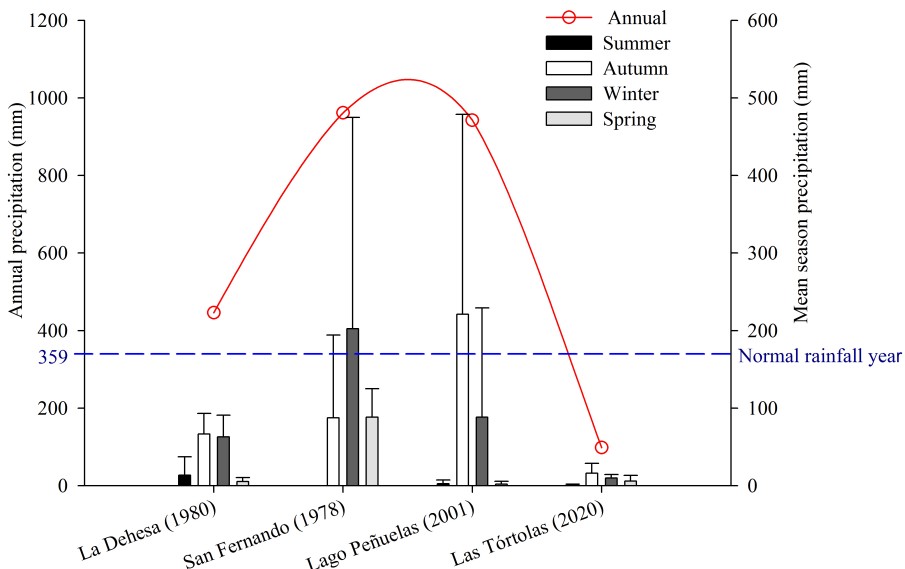

**Figure 2** Comparison of average precipitation (seasonal and annual) of the following places: La De-hesa (1980), San Fernando (1978), Lago Peñuelas (2001) y Las Tórtolas (2020, present study).

specific reference for small mammals (*Reise, 1973*; *Pearson, 1995*; *Fernández et al., 2011*; *Udrizar Sauthier et al., 2020*), birds (*Araya et al., 1995*), arthropods (*Peña, 1986*; *Artigas, 1994a*; *Artigas, 1994b*; *Aguilera & Casanueva, 2005*; *Vidal & Guerrero, 2007*). In addition, the reference collection of the Zoology Museum of the University of Concepción, Chile (MZUC-UCCC) was also reviewed.

Prey quantification was estimated using the Minimum Number of Individuals (MNI) for diagnostic structures. For vertebrates, it is based on the number of skulls and pair of jaws (*Pearson, 1995*), whereas for arthropods it is based on the presence of cranial capsules, telson and mandibles (*Martínez, 2018*). Chitinous remains in good condition were taxonomically determined using specialised references and keys (mentioned in the previous paragraph) and compared with the museum collections mentioned above. This method is thought to minimise estimation biases by aggregation (*McDowell & Medlin, 2009*). The field collections were authorised under permits from Servicio Agrícola Ganadero, SAG, of the Chilean government (No 4141/2018 and 1646/2019).

## Morphometry and hypervolume

To evaluate morphology pellet differences between the seasons, first we develop a one-way analysis of variance (ANOVA), verifying the assumptions of normal distribution and homogeneity of variance of the data (*Zar, 1999*). When a significant result was given, a Tukey's test (HSD) for multiple pairwise comparisons were performed. Second, we used a hypervolume estimation (*Blonder et al., 2014*; *Blonder et al., 2017*) to describe pellet characteristics and overlaps (shape, size and weight). This method allows the description of the shape, volume and overlap of features, attributes or variables using a Gaussian Kernel density estimation (*Blonder et al., 2017*). To determine the superposition of two

hypervolumes (autumn *vs* winter, and spring *vs* summer) similarity indices of Jaccard and Sorensen were calculated, as well as the unique fractions of the hypervolumes (*Blonder et al., 2017*). The morphometric variables of the pellets were standardised by a log-transformation. The analysis was performed using the Hypervolume package (*Blonder et al., 2017*) in the R software (*R Core Team, 2021*).

### Diet composition

To described the diet composition we used two estimators; frequency of occurrence of each prey unit (%F), and the percentage of biomass (%B). Formulas in Supplementary Materials S1.

### Characterization of the diet and trophic niche

The diet composition of *B. magellanicus* was described at two different time scales: annual and seasonal. The specific richness or the number of species in the diet (S) was determined as species count per site, and the diversity of the diet was estimated following the Shannon-Wiener index (H'), that establishes the relationship between the species richness and the abundance of each species. This index normally fluctuates between 0.5 and 5.0, where values over 3.0 are considered high levels of diversity. Additionally, we used the Pielou index ($J'$), which measures the evenness of the observed diversity. The values range from 0 to 1, with 1 corresponding to a situation where all species are equally abundant and 0 indicating the absence of uniformity (*Magurran, 1998*).

The trophic niche breadth was evaluated using the Levins (*B*) and standardised Levins ($B_{STA}$) indices (*Levin, 1968*). Levins' index is the inverse of Simpson's index. It reaches its maximum when the frequencies are the same for each prey unit consumed, and its minimum when only one resource is consumed, *i.e.,* being a strict specialist (*Krebs, 1989*). All formulas are available in Supplementary Materials S1.

### Surrounding small mammals and diet selectivity

The estimation of the richness and relative abundance of small mammals in the surrounding area was quantified through a catch per season developed from 2018 to 2020, as a complementary study that has been carried out in the surroundings of the perch (pellets point), with an annual installation of 2,800 Sherman traps (700 traps per season) and a low capture efficiency of 4.5%. The estimation of the richness and relative abundance of small mammals in the surrounding area was developed from 2018 to 2020 in a complementary study carried out in the same place, with an annual installation of 2,800 Sherman traps (700 traps per season). However, for the present study we only considered the traps installed in a surrounding radius of 700 m2 of the perch (60 traps per season, adding up to 240 traps per year with a total sampling effort equivalent to 960 traps, as each trap was active during two consecutive nights), as a potential feeding area of the owl. Each captured individual was identified at the species level, following *Reise (1973)*, *Muñoz Pedreros, Rau & Yánez (2004)* and *Muñoz Pedreros & Gil (2009)*. In the case of arthropods, study was carried out in summer (2018) and autumn (2020) using Barber soil traps, so it was not considered for the diet selectivity analysis, due to the lack of information for the remaining seasons (winter-spring).

The diet selectivity analysis was defined by a Chi-square goodness of fit test (*Sokal & Rohlf, 1969*), where the observed values correspond to the absolute frequencies of prey species occurrence in the pellets, whereas the expected values were calculated as the relative frequency of each prey species in the trapping study, matched up with the grand total of prey individuals detected in the pellets (*Jaksic & Yáñez, 1980*).

## Comparative diet analysis

The collection of pellets was done during a climatic period defined as Mega-Drought (MD) in central Chile (*Garreaud et al., 2020*), in which the year 2019-2020 was characterised as one of the driest on record with a precipitation deficit of 72.6% with respect to a normal year, only precipitating at winter time a total of 29.90 mm (information record from the environmental Huechún station, Metropolitan Region) (Fig. 2). Under this extreme environmental scenario, a comparison with other studies on diet and trophic ecology of this species in the Mediterranean zone of Chile was assembled. The diet comparison was carried out using the following studies: *Muñoz Pedreros et al. (2017)* (annual study) in the Lago Peñuelas National Reserve, Valparaíso region; *Jaksic & Yáñez (1980)* (spring study) developed in La Dehesa sector, north of Santiago, Metropolitan region; and *Yáñez, Rau & Jaksic (1978)* (summer study) carried out in the city of San Fernando, O'Higgins region.

To test the null hypothesis that there are no differences in the relative frequencies of PU between the study periods, a multivariate analysis of variance, based on permutations (PERMANOVA) of three factors (*Anderson, 2001*) was applied. The data was based on Bray-Curtis dissimilarity, squared root transformed and 999 permutations were performed. The analyses were carried out through the vegan package (*Oksanen et al., 2020*) in R software (*R Core Team, 2021*). Data and script are available in supplementary material and through the Figshare repository (https://doi.org/10.6084/m9.figshare.20481486.v2).

# RESULTS

## Morphometry and hypervolume pellets

A total of 120 pellets were analysed during a year of study (2019-2020), establishing morphometric variations between the different seasons (Table 1), with significant differences in weight (ANOVA $F = 17.12$; $p < 0.05$) and length (ANOVA $F = 2.78$; $p = 0.04$, Table 2). The Tukey's post hoc test determined difference of weight between autumn-winter, winter-spring and winter-summer for lengh, and difference of length between autumn-winter for length, while the width of the pellet did not show significant differences between the different periods (ANOVA $F = 0.39$; $p = 0.75$) (Table 2). The hypervolumes exhibited the highest volume in winter (0.75), followed by spring (0.55); whereas in summer (0.48) and autumn (0.24) the hypervolumes showed smaller sizes (Fig. 3). Regarding the superposition of the hypervolumes between seasons, a high similarity was established in terms of morphology and weight between the spring-summer and autumn-winter (Table 3).

## Characterization of the diet and trophic niche

The specific richness of diet consisted in 22 species (prey units) with a marked seasonal fluctuation along the year, registering the highest value in winter ($S = 14$) and the lowest in

**Table 1** Average values of the different morphological variables measured in the pellets, by each sampling period.

| Season | Weight | Length | Width |
|---|---|---|---|
| Autumn | $6.24 \pm 1.29$ | $5.91 \pm 0.73$ | $2.42 \pm 0.25$ |
| Winter | $41.50 \pm 1.31$ | $4.56 \pm 1.01$ | $2.40 \pm 0.37$ |
| Spring | $6.14 \pm 1,76$ | $4.92 \pm 1.03$ | $2.45 \pm 0.40$ |
| Summer | $5.36 \pm 1.11$ | $4.89 \pm 0.97$ | $2.36 \pm 0.35$ |

**Table 2** One-way ANOVA analysis of each morphological and weight variable of the pellets.

| Variable | Sum of squares | df | Mean square | F | p |
|---|---|---|---|---|---|
| **Weight** | | | | | |
| Between groups: | 0.687 | 3 | 0.229 | 17.120 | 0.000[*] |
| Within groups: | 1.550 | 116 | 0.013 | | |
| Total: | 2.237 | 119 | 0.000 | | |
| **Large** | | | | | |
| Between groups: | 0.054 | 3 | 0.018 | 2.78 | 0.04[*] |
| Within groups: | 0.757 | 116 | 0.007 | | |
| Total: | 0.811 | 119 | 0.043 | | |
| **Width** | | | | | |
| Between groups: | 0.004 | 3 | 0.001 | 0.399 | 0.754[n.s.] |
| Within groups: | 0.409 | 116 | 0.004 | | |
| Total: | 0.414 | 119 | 0.756 | | |

**Notes.**
[*]$p < 0.05$ ; n.s.= no significant.

spring ($S = 6$). The Shannon-Wiener index showed an average of $0.71 \pm 0.33$ bit/ind., with lower values in the spring (0.35 bit/ind.) and summer (0.53 bit/ind.), contrasting what was exhibited in autumn (1,086 bit/ind.) and winter (0.89 bit/ind.). The Pielou index showed the same trend as the specific diversity, with greater heterogeneity of diet in autumn (0.44), and lower evenness in the rest of the seasons (winter = 0.17; spring = 0.19; summer = 0.23) due to the dominance of a few PU (Table 4). The annual estimation of the trophic niche breadth was $B = 0.37$, while the standardised Levin's index was $B_{STA} = 0.01$.

We quantified 1,071 annual PU and consumed biomass of 15,988 (g per year). The higher PU percentages identified in the pellets occurred in summer (34.30%; PU = 367) and spring (28.70%; PU = 307), adding up to 63% (PU = 674) of the total PU. The percentages of consumed biomass showed the same tendency, with higher values in summer (32.36%; 5,174 g) and spring (26.85%; 4,293 g). Most of the annual PU consumed (97.75%) were arthropods (87.86% Arachnida and 9.89% Insecta); and the remaining 2.24% were vertebrates. *Grammostola rosea* (from the class Arachnida) was the most consumed species with an annual PU percentage of 84.97% (PU = 910), plus it exhibited the highest percentage of occurrence throughout the year ($77.48 \pm 13.22\%$ PU), with maximum values in spring (92.73% of consumed biomass) and summer (88.63% PU). The scorpion of the genus *Brachistosternus* was scarcely consumed (2.89% annual PU), showing a low percentage of occurrence throughout the year ($0.14 \pm 0.09\%$) with maximum values

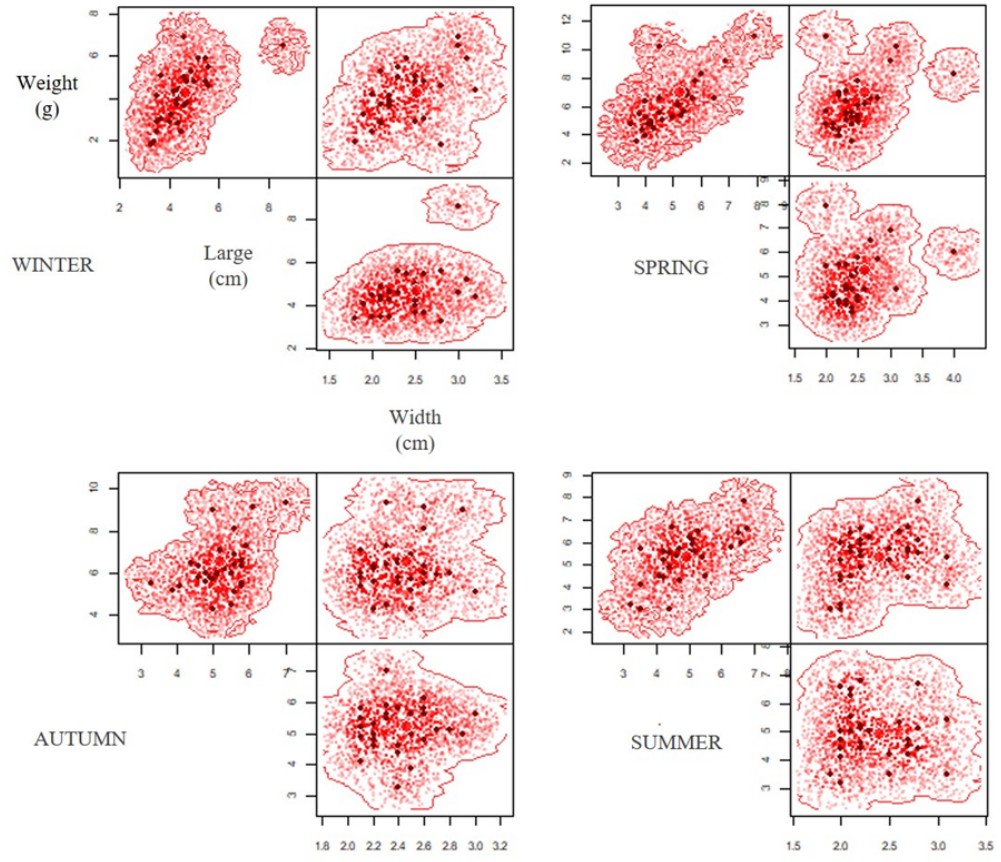

**Figure 3** Seasonal variation of hypervolume dimension (width, large and weight) of pellets during the period of study (extreme drought 2019–2020).

**Table 3 Similarity indices and unique fractions for hypervolumes by the different seasons of the year.** Unique fraction 1 = volume of unique component of 1 divided by volume of 1 and Unique fraction 2 = volume of unique component of 2 divided by volume of 2.

| Seasons | Jaccard | Sorensen | Unique fraction 1 | Unique fraction 2 |
|---|---|---|---|---|
| Autumn *vs* Winter | 0.254 | 0.405 | 0.176 | 0.732 |
| Spring *vs* Summer | 0.463 | 0.633 | 0.402 | 0.327 |

of consumed biomass in autumn (0.27%). Among Insecta, Tenebrionidae family was the most consumed with 8.40% annual PU and with an annual biomass contribution of $0.23 \pm 0.11\%$.

Vertebrates were poorly represented in the diet, with only 2.24% of the annual PU and a marked seasonal occurrence percentage, with a maximum in autumn (33.33% PU) and a minimum in spring (3.33% PU). The annual average of biomass consumed of this group was $21.67 \pm 15.74\%$, with the lowest values recorded in spring (5.27%) The rodents

**Table 4 Diet composition of *Bubo magellanicus* in different seasons in a Mediterranean scrub habitat (33°9′41.50″S, 70°42′23.25″W).**

| Habitat | | Mediterranean scrub | | | | | | | | | | | |
| Season | | Autumn | | | Winter | | | Spring | | | Summer | | |
| Date | | 4-14 June | | | 2-12 August | | | 6-17 October | | | 3-14 January | | |
| No pellets | | 30 | | | 30 | | | 30 | | | 30 | | |
| Prey Units | | 229 | | | 167 | | | 307 | | | 367 | | |
| Category | Mass (g) | No | %F | %B | No | %F | %B | No | %F | %B | No | %F | %B |
|---|---|---|---|---|---|---|---|---|---|---|---|---|---|
| **Total Mammalia** | | 10 | 33.33 | 36.82 | 8 | 26.67 | 33.37 | 1 | 3.33 | 5.27 | 4 | 13.33 | 11.18 |
| *Phyllotis darwini* | 61.65 | 2 | 6.67 | 3.39 | 2 | 6.67 | 4.27 | 0 | 0.00 | 0.00 | 1 | 3.33 | 1.19 |
| *Abrocoma bennetti* | 226.25 | 1 | 3.33 | 6.23 | 2 | 6.67 | 15.66 | 1 | 3.33 | 5.27 | 2 | 6.67 | 8.75 |
| *Abrothrix longipilis* | 64.25 | 2 | 6.67 | 3.54 | 3 | 10.00 | 6.67 | 0 | 0.00 | 0.00 | 1 | 3.33 | 1.24 |
| *Abrothrix olivaceus* | 34.95 | 2 | 6.67 | 1.92 | 0 | 0.00 | 0.00 | 0 | 0.00 | 0.00 | 0 | 0.00 | 0.00 |
| *Octodon degus* | 195.3 | 0 | 0.00 | 0.00 | 1 | 3.33 | 6.76 | 0 | 0.00 | 0.00 | 0 | 0.00 | 0.00 |
| *Spalacopus cyanus* | 87 | 1 | 3.33 | 2.39 | 0 | 0.00 | 0.00 | 0 | 0.00 | 0.00 | 0 | 0.00 | 0.00 |
| *Oryctolagus cuniculus* | 667 | 1 | 3.33 | 18.36 | 0 | 0.00 | 0.00 | 0 | 0.00 | 0.00 | 0 | 0.00 | 0.00 |
| *Thylamys elegans* | 35.60 | 1 | 3.33 | 0.98 | 0 | 0.00 | 0.00 | 0 | 0.00 | 0.00 | 0 | 0.00 | 0.00 |
| **Total Birds** | | 0 | 0.00 | 0.00 | 0 | 0.00 | 0.00 | 1 | 3.33 | 1.86 | 0 | 0.00 | 0.00 |
| *Turdus cf falcklandii* | 80 | 0 | 0.00 | 0.00 | 0 | 0.00 | 0.00 | 1 | 3.33 | 1.86 | 0 | 0.00 | 0.00 |
| **Total Insects** | | 44 | 50.00 | 0.48 | 21 | 43.33 | 0.21 | 15 | 23.33 | 0.13 | 26 | 46.67 | 0.20 |
| Tettigonidae/Sp1 | 0.80 | 0 | 0.00 | 0.00 | 0 | 0.00 | 0.00 | 1 | 3.33 | 0.02 | 0 | 0.00 | 0.00 |
| *Arthrobrachus maestus* | 0.10 | 0 | 0.00 | 0.00 | 1 | 3.33 | 0.00 | 0 | 0.00 | 0.00 | 0 | 0.00 | 0.00 |
| *Proacis* sp. | 0.35 | 28 | 26.67 | 0.27 | 5 | 6.67 | 0.06 | 14 | 20.00 | 0.11 | 13 | 16.67 | 0.09 |
| *Nycterinus* sp. | 0.40 | 0 | 0.00 | 0.00 | 1 | 3.33 | 0.01 | 0 | 0.00 | 0.00 | 0 | 0.00 | 0.00 |
| Tenebrionidae/Sp1 | 0.50 | 15 | 20.00 | 0.21 | 5 | 10.00 | 0.09 | 0 | 0.00 | 0.00 | 10 | 20.00 | 0.10 |
| Tenebrionidae/Sp2 | 0.10 | 0 | 0.00 | 0.00 | 2 | 6.67 | 0.01 | 0 | 0.00 | 0.00 | 0 | 0.00 | 0.00 |
| Tenebrionidae/Sp3 | 0.10 | 0 | 0.00 | 0.00 | 4 | 6.67 | 0.01 | 0 | 0.00 | 0.00 | 0 | 0.00 | 0.00 |
| Tenebrionidae/Sp4 | 0.35 | 0 | 0.00 | 0.00 | 1 | 3.33 | 0.01 | 0 | 0.00 | 0.00 | 0 | 0.00 | 0.00 |
| Carabidae/Sp1 | 0.20 | 0 | 0.00 | 0.00 | 2 | 3.33 | 0.01 | 0 | 0.00 | 0.00 | 1 | 3.33 | 0.00 |
| Coleotera/Sp1 | 0.10 | 1 | 3.33 | 0.00 | 0 | 0.00 | 0.00 | 0 | 0.00 | 0.00 | 1 | 3.33 | 0.00 |
| Hymenoptera/Sp1 | 0.25 | 0 | 0.00 | 0.00 | 0 | 0.00 | 0.00 | 0 | 0.00 | 0.00 | 1 | 3.33 | 0.00 |
| **Total Arachnida** | | 175 | 100.00 | 62.70 | 138 | 90.00 | 66.42 | 291 | 113.33 | 92.73 | 337 | 116.67 | 88.63 |
| *Grammostola rosea* | 14 | 162 | 76.67 | 62.43 | 137 | 86.67 | 66.40 | 284 | 96.67 | 92.61 | 327 | 93.33 | 88.48 |
| *Brachistosternus* sp. | 0.75 | 13 | 23.33 | 0.27 | 1 | 3.33 | 0.03 | 7 | 16.67 | 0.12 | 10 | 23.33 | 0.14 |
| TOTAL MASS | | 3,632 | | | 2,888 | | | 4,293 | | | 5,174 | | |

Notes.

%F, percentage of frequency; %B, percentage of biomass.

*Abrocoma bennetti* and *Abrothrix longipilis* were the most consumed prey with 52.15% of vertebrate annual PU, although it only represents 1.12% of the total annual PU. The marsupial *Thylamys elegans* and the bird *Turdus falcklandii* were scarcely consumed with only one record in autumn and spring, respectively (Table 4).

**Table 5  Catch number and relative abundance of small mammals found in pellets and trapped by Sherman traps in the surrounding area ($X^2 = 7.817$; $p < 0.05$).**

| Species | Autumn | | Winter | | Spring | | Summer | | $X^2$ |
|---|---|---|---|---|---|---|---|---|---|
| | No capture | Relative abundance | No capture | Relative abundance | No capture | Relative abundance | No capture | Relative abundance | |
| *Phyllotis darwini* | 3 | 5 | 2 | 3.33 | 1 | 1.67 | 2 | 3.33 | 0.89 |
| *Abrothrix longipilis* | 0 | 0.00 | 0 | 0.00 | 1 | 1.67 | 1 | 1.67 | 5.33 |
| *Abrothrix olivaceus* | 1 | 1.67 | 1 | 1.67 | 1 | 1.67 | 1 | 1.67 | 3.01 |
| *Thylamys elegans* | 1 | 1.67 | 0 | 0.00 | 1 | 1.67 | 0 | 0.00 | 0.81 |

## Surrounding small mammals and diet selectivity

A low richness and relative abundance of small mammals was recorded during the different sampling seasons. *Phyllotis darwini* was the most captured rodent with a mean relative abundance of $3.33 \pm 1.36\%$, while the rest of mammals presented relative abundances less than or equal to 1% (Table 5). The selectivity of the diet established that *B. magellanicus* did not consume mammals in the same proportion as they are present in the surrounding area ($X^2 = 7.81$; $p < 0.05$). In decreasing order, the most consumed rodents were *A. bennetti*. *A. longipilis* and *P. darwini*, whereas *Octodon degus* and *Spalacopus cyanus* were the least consumed. The marsupial *T. elegans* and the lagomorph *Oryctolagus cuniculus* were extremely rare in the diet (See Table 4).

## Comparative diet analysis

In comparison to other studies on dietary ecology of *B. magellanicus* performed in the Mediterranean region of Chile, this study determined the lowest trophic niche breadth record for the species, with values of Levin's index $B = 1.38$ and Levins standardised index $B_{STA} = 0.01$ (Lago Peñuelas $B = 6.23$; $B_{STA} = 0.37$. La Dehesa $B = 6.32$; $B_{STA} = 0.66$. San Fernando $B = 2.93$; $B_{STA} = 0.18$). The Shannon-Wiener and Pielou indices followed the same trend, with the lowest values estimated for the species, whereas the dominance index and prey richness exhibited the highest values compared to the other trophic studies (Table 6).

Trophic characterization showed a shift in the composition and biomass of the diet for *B. magellanicus* during the period of extreme drought, exhibiting preferences of arthropods over vertebrates. In general, most of the biomass consumed (78%) corresponded to arthropods, with vertebrates contributing the resting 22%, thus contrasting the results of Lago Peñuelas National Reserve (*Muñoz Pedreros et al., 2017*), that described high preference for small mammals ($90.17 \pm 10.33\%$ diet biomass) and a marginal percentage consumption in arthropods (<6.0% PU). Also, the results of La Dehesa study, carried out only in spring (*Jaksic & Yáñez, 1980*) described consumption only for vertebrates (small mammals 88.7% and birds 11.3%) compared to 66.6% consumption of arthropods biomass (0.2% Insecta and 66.4% Arachnida) and only the remainning 33.4% of vertebrates biomass, in the same season. Nevertheless, the research developed in summer in San Fernando (*Yáñez, Rau & Jaksic, 1978*) showed a greater consumption of arthropods than the rest of the comparative studies (17.70%), but less than the percentages of consumption registered

**Table 6 Comparison of the composition of *Bubo magellanicus* diet in Mediterranean environments of central Chile.**

| Habitat | | | Mediterranean scrub | | | | | |
| Locations | Las Tórtolas-Colina | | NR Lago Peñuelas | | La Dehesa (Spring) | | San Fernando (Summer) | |
| Latitude | 33°9′S | | 33°9′S | | 33°21′S | | 34°35′S | |
| Longitude | 70°42′W | | 71°31′W | | 70°32′W | | 70°59′W | |
| Category PU | F% | B% | F% | B% | F% | B% | F% | B% |
|---|---|---|---|---|---|---|---|---|
| **Total Mammalia** | 16.70 | 21.70 | 69.00 | 95.00 | 88.70 | 97.20 | 28.80 | 78.30 |
| *Thylamys elegans* | 0.80 | 0.20 | 2.20 | 0.40 | 3.50 | 0.60 | 0.00 | 0.00 |
| *Oligoryzomys longicaudatus* | 0.00 | 0.00 | 7.10 | 1.20 | 4.40 | 0.70 | 2.20 | 1.50 |
| *Abrothrix longipilis* | 4.40 | 2.90 | 5.70 | 1.60 | 16.70 | 4.90 | 0.00 | 0.00 |
| *Abrothrix olivaceus* | 1.70 | 0.50 | 7.30 | 1.20 | 0.90 | 0.10 | 0.40 | 0.30 |
| *Chelemys megalonyx* | 0.00 | 0.00 | 0.00 | 0.00 | 0.00 | 0.00 | 0.40 | 0.50 |
| *Phyllotis darwini* | 3.60 | 2.20 | 0.80 | 0.30 | 4.40 | 1.40 | 6.10 | 9.80 |
| *Loxodontomys micropus* | 0.00 | 0.00 | 0.00 | 0.00 | 0.00 | 0.00 | 1.30 | 1.90 |
| *Octodon degus* | 0.80 | 1.70 | 0.30 | 0.20 | 0.00 | 0.00 | 0.00 | 0.00 |
| *Octodon lunatus* | 0.00 | 0.00 | 2.70 | 2.10 | 0.00 | 0.00 | 0.00 | 0.00 |
| *Spalacopus cyanus* | 0.80 | 0.60 | 0.80 | 0.40 | 0.00 | 0.00 | 0.00 | 0.00 |
| *Abrocoma bennetti* | 3.80 | 9.00 | 18.20 | 23.00 | 18.40 | 24.10 | 0.00 | 0.00 |
| *Rattus norvegicus* | 0.00 | 0.00 | 0.00 | 0.00 | 0.00 | 0.00 | 7.00 | 14.50 |
| *Rattus rattus* | 0.00 | 0.00 | 1.10 | 0.90 | 19.30 | 17.10 | 0.90 | 3.10 |
| *Mus musculus* | 0.00 | 0.00 | 0.80 | 0.10 | 0.00 | 0.00 | 0.00 | 0.00 |
| Undetermined rodents | 0.00 | 0.00 | 0.00 | 0.00 | 5.30 | 1.00 | 7.00 | 5.30 |
| *Lepus europeaus* | 0.00 | 0.00 | 0.00 | 0.00 | 0.00 | 0.00 | 0.00 | 0.00 |
| *Oryctolagus cuniculus* | 0.80 | 4.60 | 220 | 63.60 | 15.80 | 47.30 | 3.50 | 41.40 |
| **Total Birds** | **0.80** | **0.50** | **23.60** | **5.10** | **11.30** | **2.80** | **6.90** | **6.10** |
| Passeriformes | 0.00 | 0.00 | 0.00 | 0.00 | 11.30 | 2.80 | 1.70 | 1.50 |
| Undetermined aves | 0.00 | 0.00 | 23.60 | 5.10 | 0.00 | 0.00 | 5.20 | 4.60 |
| *Turdus cf falcklandii* | 0.80 | 0.50 | 0.00 | 0.00 | 0.00 | 0.00 | 0.00 | 0.00 |
| **Total Insects** | **35.70** | **0.30** | **7.30** | **0.00** | **0.00** | **0.00** | **12.20** | **0.30** |
| Coleoptera | 1.10 | 0.00 | 5.20 | 0.00 | 0.00 | 0.00 | 4.80 | 0.10 |
| Tettionidae/Sp1 | 0.80 | 0.00 | 0.00 | 0.00 | 0.00 | 0.00 | 0.00 | 0.00 |
| *Arthrobrachus maestus* | 0.80 | 0.00 | 0.00 | 0.00 | 0.00 | 0.00 | 0.00 | 0.00 |
| *Proacis* sp. | 16.60 | 0.10 | 0.00 | 0.00 | 0.00 | 0.00 | 0.00 | 0.00 |
| *Nycterinus* sp. | 0.80 | 0.00 | 0.00 | 0.00 | 0.00 | 0.00 | 0.00 | 0.00 |
| Tenebrioniidae/Sp1 | 10.00 | 0.10 | 0.00 | 0.00 | 0.00 | 0.00 | 0.00 | 0.00 |
| Tenebrioniidae/Sp2 | 1.70 | 0.00 | 0.00 | 0.00 | 0.00 | 0.00 | 0.00 | 0.00 |
| Tenebrioniidae/Sp3 | 1.70 | 0.00 | 0.00 | 0.00 | 0.00 | 0.00 | 0.00 | 0.00 |
| Tenebrioniidae/Sp4 | 0.80 | 0.00 | 0.00 | 0.00 | 0.00 | 0.00 | 0.00 | 0.00 |
| Carabidae | 1.10 | 0.00 | 0.00 | 0.00 | 0.00 | 0.00 | 0.00 | 0.00 |
| Hymenoptera | 0.30 | 0.00 | 0.00 | 0.00 | 0.00 | 0.00 | 0.00 | 0.00 |
| Orthoptera | 0.00 | 0.00 | 0.50 | 0.00 | 0.00 | 0.00 | 7.40 | 0.20 |
| Undetermined Insecta | 0.00 | 0.00 | 1.60 | 0.00 | 0.00 | 0.00 | 0.00 | 0.00 |
**Table 6** (*continued*)

| Habitat<br>Locations<br>Latitude<br>Longitude | Mediterranean scrub | | | | | | | |
|---|---|---|---|---|---|---|---|---|
| | Las Tórtolas-Colina<br>33°9′S<br>70°42′W | | NR Lago Peñuelas<br>33°9′S<br>71°31′W | | La Dehesa (Spring)<br>33°21′S<br>70°32′W | | San Fernando (Summer)<br>34°35′S<br>70°59′W | |
| **Category PU** | **F%** | **B%** | **F%** | **B%** | **F%** | **B%** | **F%** | **B%** |
| **Total Arachnida** | **160.10** | **77.60** | **0.00** | **0.00** | **0.00** | **0.00** | **52.20** | **17.40** |
| *Grammostola* sp. | 0.00 | 0.00 | 0.00 | 0.00 | 0.00 | 0.00 | 47.40 | 15.80 |
| *Grammostola rosea* | 146.80 | 77.50 | 0.00 | 0.00 | 0.00 | 0.00 | 0.00 | 0.00 |
| *Brachistosternus* sp. | 13.30 | 0.10 | 0.00 | 0.00 | 0.00 | 0.00 | 0.00 | 0.00 |
| Scorpionida | 0.00 | 0.00 | 0.00 | 0.00 | 0.00 | 0.00 | 4.80 | 1.60 |
| Total | 1071 | | 368 | | 114 | | 230 | |
| Pellets | 120 | | 241 | | 98 | | 63 | |
| Levins index B | 1.38 | | 6.23 | | 6.32 | | 2.93 | |
| Levins standar $B_{STA}$ | 0.01 | | 0.37 | | 0.66 | | 0.18 | |
| Richness | 22 | | 16 | | 10 | | 15 | |
| Shannon-Wiener index | 0.72 | | 2.14 | | 2.06 | | 1.94 | |
| Dominance index | 0.73 | | 0.15 | | 0.14 | | 0.25 | |
| Pielou index | 0.23 | | 0.77 | | 0.89 | | 0.72 | |

**Notes.**

F%, percentage of frequency; B%, percentage of biomass.

in our study (88.8%). However, the present study exhibited significant differences of diet composition with all the trophic ecology studies previously documented for the species in the Mediterranean region of Chile (PERMANOVA $F = 2.59$; $p < 0.05$).

The arachnid *G. rosea* was the species that most contributed annually to the biomass (77.50%), followed by the rodent *A. bennetti* (9.01%) in this study, contrasting with *Jaksic & Yáñez (1980)* and *Muñoz Pedreros et al. (2017)* which determined an average contribution of 50.76 ± 11.76% by *O. cuniculus* and 23.55 ± 0.78% by *A. bennetti*, respectively. The study of *Yáñez, Rau & Jaksic (1978)* described an important contribution from the genus *Grammostola* sp. (15.80%) and the exotic rodent *Rattus norvegicus* (14.50%).

# DISCUSSION

The variation in size and weight of the pellets have been directly related to the availability and type of prey, as well as the abundance of the surrounding vertebrates. In turn, the abundance of prey in the environment would depend, among other factors, on the presence and availability of resources, which fluctuate throughout the seasons (*Schlatter et al., 1982*; *Jaksic, Yáñez & Rau, 1986*; *Tala, González & Bonacic, 1995*; *Nabte, Saba & Pardiñas, 2006*; *Mella et al., 2016*). Other studies of the diet of *B. magellanicus* (*Mella et al., 2016*), *Athene cunicularia* (*Schlatter et al., 1982*; *Vieira & Teixeira, 2008*) and others owl (*Comay & Dayan, 2018*) have determined seasonal variations in the size of pellets, with both showing minimum values in spring and maximum values in autumn and winter, respectively. This morpho-variation in pellets is similar to what was reported in this study, with higher hypervolume values in winter (Vol = 0.75) and spring (Vol = 0.55) (see Fig. 3). According to *Schlatter et al. (1982)* these differences in size are explained in part by the greater

proportion of small mammals in the diet and the greater consumption of adult individuals, which agrees with what was recorded by *Nabte, Saba & Pardiñas (2006)*. However, the low proportion of vertebrates consumption determined in this study is possibly related to the intake of adults of the mygalomorph spider *G. rosea*, which has greater availability in the field with nocturnal nomadic behaviours (*Grossi et al., 2016*; *Aguilera, Montenegro & Casanueva, 2019*).

Diet composition of *B. magellanicus* also showed seasonal variation probably influenced by the presence and diversity of prey in the environment (*Nabte, Saba & Pardiñas, 2006*; *Muñoz Pedreros et al., 2017*; *Comay & Dayan, 2018*). An example of this, is the number of prey units (PU) and biomass quantified in spring (308 PU; 4,293 g) and summer (367 PU; 5,174 g), which would respond to a greater supply of male representatives of *G. rosea* during their period of reproduction (*Aguilera, Montenegro & Casanueva, 2019*) or juveniles during terrestrial dispersal events that occur between December and March (*Montenegro-Vargas, Montenegro-Heidke & Aguilera, 2022*). The high richness of this taxon in the diet of *B. magellanicus* was due to the exhaustive taxonomic identification of invertebrates, which revealed a high participation in its composition and biomass in comparison to other studies of diet of this owl in Chile (*Yáñez, Rau & Jaksic, 1978*; *Jaksic & Yáñez, 1980*; *Jaksic, Yáñez & Rau, 1986*; *Tala, González & Bonacic, 1995*; *Mella et al., 2016*; *Muñoz Pedreros et al., 2017*; *Martínez, 2018*; *Vega, Jara & Mella, 2018*; *Zuñiga et al., 2022*). Nevertheless, a low diversity diet was determined due to an almost exclusive consumption of invertebrates (97.75%), specifically *G. rosea* (87.86%), thus obtaining the lowest value of trophic niche breadth described in *B. magellanicus* in a Mediterranean habitat ($B = 0.37$; $B_{STA} = 0.01$) (*Yáñez, Rau & Jaksic, 1978*; *Jaksic & Yáñez, 1980*; *Muñoz Pedreros et al., 2017*). A similar feeding preference was reported by Udrizar Sauthier et al. (2017a) at an insular system off Argentine Patagonia, with over 70% of invertebrates contributing to diet biomass due to the isolation of the study area and the absence of the usual prey species on the island, especially native rodents (*Udrizar Sauthier et al., 2017b*). But our study was performed in the central of Chile, a region considered as one of the 25 diversity hotspots on Earth (*Myers et al., 2000*) sustaining over 50% of Chile's vertebrate species, 50% of Chilean endemic species, and 50% of the endangered species (*Simonetti, 1999*). Therefore, the low presence of small mammals in the study area due to isolation was discarded as a possibility, and instead could be suggested to be related to a climatic scenario of extreme drought that has developed more than a decade in the Mediterranean region of Chile (*Garreaud et al., 2020*), affecting the hydroclimate and vegetation plus causing complex changes in composition and productivity (*Garreaud et al., 2017*), that generates trophic cascades' modifications due down-top control (*Elmhagen & Rushton, 2007*; *Lesser et al., 2020*) that in turn transforms the structure and composition of the community, promoting long-term persistence of rare species by stressing dominant species throughout the food web (*Prugh et al., 2018*; *De Necker et al., 2022*). The trophic ecology of owls is also affected by the alteration of energy transfer flux through the different links that end up affecting the strong trophic interactions present in the food web (*McCann, Hastings & Huxel, 1998*; *Prugh et al., 2018*). To cope with these extreme periods, consumers would increase their dependence on weak interactions to stabilise dynamics and absorb energy in order to increase their trophic niche

(*Van Valen, 1965*; *McCann, Hastings & Huxel, 1998*; *Pool et al., 2017*; *Prugh et al., 2018*), or would strengthen their dependence on some strong links, to optimise their foraging in energetically favourable pathways, narrowing their niche size (*MacArthur & Pianka, 1966*; *Pyke, Pulliam & Charnov, 1977*; *Cachera et al., 2017*), depending on the availability of prey on the environment (*Werner & Hall, 1974*; *Iwasa, Higashi & Yamamura, 1981*; *Comay & Dayan, 2018*). The trophic restructuration that we report in this study is a decrease in trophic niche breadth and a strengthening of predator–prey interaction by the specialisation in the consumption of *G. rosea*, according to the availability in the environment and the potential nutritional value, by changing the feeding strategy based on small mammals (*Rau & Yáñez, 1981*; *Tala, González & Bonacic, 1995*; *Mella et al., 2016*; *Muñoz Pedreros et al., 2017*; *Martínez, 2018*; *Vega, Jara & Mella, 2018*) with some bias in prey biomass (>20 g) (*Jaksic & Marti, 1984*) to an exclusive specialist of small invertebrates (<20 g). These changes in the selection of prey in avian raptors are caused and influenced by the scarcity of their usual prey due to drastic decreases in their populations (*Heywood & Pavey, 2002*; *Comay & Dayan, 2018*) as a result of extreme climatic scenarios, such as the mega-drought reported for the region (*Garreaud et al., 2017*; *Garreaud et al., 2020*; *McDowell & Medlin, 2009*). Nonetheless, not only their diet is affected, as there may also be possible drastic effects on their populations, reproduction and biomass of individuals (*Cruz-McDonnell & Wolf, 2016*; *Fromant et al., 2021*).

Diet selectivity in owls is still under discussion, as some authors report a relationship between the proportion of prey consumed and its availability in the field (*Jaksic & Yáñez, 1980*; *Mella et al., 2016*), whereas other authors do not find an empiric relationship between consumption and field prey proportion (*Muñoz Pedreros et al., 2017*). In this study, selectivity was only analysed among small mammals, due to the complementary trapping study and the synchrony with the collection of pellets, finding no apparent relationship between the proportions of consumed and trapped individuals. However, despite the lack of consensus on this matter, pellets have been proposed as a useful estimator of abundance of small mammals (*Yom-Tov & Wool, 1997*; *McDonald, Burnett & Robinson, 2014*) and epigean arthropods (*Cheli et al., 2019*) in the environment, although with biases to be considered such as the hunting strategy, the range of prey, the diurnal-nocturnal behaviour and the hunting area of the predator (*Andrade, De Menezes & Monjeau, 2016*).

The frequency, severity and duration of droughts and heat waves will increase due to global warming (*Bellard et al., 2012*; *Satoh et al., 2022*), causing substantial changes in food webs where top predators will be the most affected, with negative effects on their reproduction and feeding. In central Chile, a long-standing drought so called MD persists, which is unprecedented within the country's historical records and has caused pressures on the physiognomy and vegetation productivity of the region, with drastic effects that have been scarcely evaluated on trophic systems.

## CONCLUSIONS

A complex environmental scenario, such as a prolonged drought, can lead to drastic changes in the trophic ecology of a top predators (*B. magellanicus*), overstraining predator–prey interactions and shifting their food preferences to lower nutritional links, as less biomass consumption (arthropods).

## ACKNOWLEDGEMENTS

We thank Pablo Fuentes for helping us in the taxonomic identification of arthropods, and Manuela Pérez-Aragón for her comments and translation in this manuscript.

### Funding

Reinaldo Rivera was supported by the Millennium Institute of Oceanography (IMO), University of Concepción, Concepción, Chile and by the ANID FONDECYT Grant (1201506). The funders had no role in study design, data collection and analysis, decision to publish, or preparation of the manuscript.

### Grant Disclosures

The following grant information was disclosed by the authors:
The Millennium Institute of Oceanography (IMO), University of Concepción, Concepción, Chile and by the ANID FONDECYT Grant: 1201506.

### Competing Interests

The authors declare there are no competing interests. Sam Catchpole Ahumada is employed by Catchpole & Alarcón, and Luis Carrera Suarez is employed by Caminando Otro Sendero E.I.R.L.

### Author Contributions

- Sam Catchpole Ahumada conceived and designed the experiments, performed the experiments, analyzed the data, prepared figures and/or tables, authored or reviewed drafts of the article, and approved the final draft.
- Luis Carrera Suárez conceived and designed the experiments, authored or reviewed drafts of the article, and approved the final draft.
- Reinaldo Rivera conceived and designed the experiments, performed the experiments, analyzed the data, prepared figures and/or tables, authored or reviewed drafts of the article, and approved the final draft.

### Field Study Permissions

The following information was supplied relating to field study approvals (i.e., approving body and any reference numbers):
The field collections were authorized by Servicio Agrícola Ganadero, SAG, of Chilean government (No 4141-2018 and No 1646-2019).

## Data Availability

The data and script are available at Figshare:

Rivera, Reinaldo (2022): Effects of an extreme drought (2019-2020) on the feeding ecology of Bubo magellanicus (Gmelin 1788) (Strigiformes: Strigidae) in a Mediterranean region of Chiletem. figshare. Dataset. https://doi.org/10.6084/m9.figshare.20481486.v2.

## Supplemental Information

Supplemental information for this article can be found online at http://dx.doi.org/10.7717/peerj.15020#supplemental-information.

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
