# Peer review of "Could an event of extreme drought (2019-2020) affect the feeding ecology of Bubo magellanicus (Gmelin 1788) (Strigiformes: Strigidae) in a Mediterranean region of Chile?"

_PeerJ, doi:10.7717/peerj.15020_

## Round 0.1 · original submission · Major Revisions

Two reviewers agree that this work should be published, but only after significant editing. One reviewer is opposed to the publishing of this text in its current form. Please revise the manuscript.

·

Basic reporting

The manuscript is very good, but it requires minor but obligatory changes, specifically add some current references from the last five years with sentences that are supported with former references.
Also, there is a weak point in statistical analysis that is verify normal distribution and/or variance homogeneity as previous condition for apply ANOVA.

Experimental design

The experiment design is very good and appropiated, but it the statistical analysis requires an obligatory condition previous to ANOVA (and for any parametric test), that is verify normal distribution and/or variance homogeneity. There is not a strict requirement about these previous condition, I suggest these options:
1- If you verify normal distribution and variance homogeneity, you must cite any classic statistical book such as Zar, J.H. (Biostatistical Analysis), or Sokhal & Rholf (Biometry)
2- If you want verify only normal distribution, you must cite papers of Aaron Ellison.
3- If you want verigy only variance homogeneity you must cite the papers of community ecology of A.J. Underwood.

Validity of the findings

The results are very valid and excellent quality, but I suggest improve the presentation including more recent references in some parts of introduction and discussion.

Additional comments

Dear authors:
Very good manuscript, it requires only minor changes previous to its publication.
Many success and blessings !!

Reviewer 2 ·

Basic reporting

The manuscript is in my opinion a very interesting study of the diet of B. magellanicus in an area affected by a mega-drought. Since I am not a native English speaker I will not refer to the writhing other than I disagree with the use of “for” in several situations in the manuscript. In my opinion, a finding is not “for” something: a percentage is not ‘for’ some owls, a record is not ‘for’ B. magellanicus; other prepositions or expressions (among, in, of, in the case of) could be used in each case. In biological science, things usually don’t happen ‘for’ something; they just happen. Some suggestion are given in the additional comments.

Experimental design

The design includes to compare the record with those of other years and locality. Since this is not a repeated study in the same locality, the evidence of an effect of the climate change is less strong that what I saw in the manuscript. In addition, the lack of studies of arthropods in the environment (only one in summer) does not support the correlations between availability of prey and consumption. And, although the word “could” is conservative, I think that “results suggest” is more adequate. In my opinion, the seeking for another explanation is lacking; for instance, in San Fernando, in a more humid region to the south of the present study, 44 years ago, vertebrates also represented a low proportion of the diet. This support that local conditions (and not a drought) could also affected the diet. I am not rejecting the hypothesis of the effect of the drought, it is very sound, but more studies could better support that (for instance, taking the opportunity that this year has been more rainy than last decade). Therefore, a more conservative discussion could improve the paper.

Validity of the findings

In spite of the above, the findings are very interesting and open an interesting research field in Chile

Additional comments

Title Page
Affiliation 4: Is it “Laboratorio de Ecología y Filoinformática, Facultad de Ciencias Naturales y Oceanográficas, Universidad de Concepción?” I apologize if I have confounded the lab.
Line (l) 36: “for B. magellanicus”; another preposition, maybe “among” or “in the species B. magellanicus”
L192: Change ‘for’ to ‘in the case of’
L208-212: these cities are to the south of the study. Please justify why they are comparable.
L229: correct to ‘…difference of the weight between..”
L230: “…, and difference of the length between the autumn-winter…”
L228-232; There is no report of the means of weight and length by season. I suggest to include these averages.
L241: Prey units: use the same meaning through the text. In this case it means presy species, in other cases (e.g. Table 3) it means prey individuals.
L245-247: The Pielou indexes should be reported here, for instance in parentheses.
L248: You cite Table 3, but Table 3 does not give the diversity indexes
L252: change “for” to “in”
L265: change “for “ to “of” (maybe “The annual average of biomass of this group consumed by B. magellanicus was 21.67…)
L275: Change “described” by “presented” or “had”
L278-280: An what are the proportions they were represented in the area? Otherwise, cite Table 3 again.
L291: change “for “ to “of”
Tables 1, 3 and 5: abbreviations must be explained: What do df, %F and %B, mean?
L297-298: The study in San Fernando in 1978 support the question: Is the low percentage of vertebrates due to climate change or local conditions? It was already mentioned at the beginning of this revision.
L305: Change sp. from italics to roman font.
L313: change “for “ to “of the diet of”
L324-325: It is not clear the difference between this sentence and those at the beginning of the previous paragraph
L331: Change “studies developed for this owl” to “studies of the diet of this owl in Chile”, or another sentence without “for”.
L336: change “for” to “in”
L345: But the study in San Fernando was more than 40 years ago and it also reported low frequency of vertebrates. Given that there is no study in the same locality some years ago, and that comparisons have been performed with other locations, I think that the extreme drought is not the only one possible explanation. In my opinion it should be a more conservative sentence, such as "results suggest that...".
L376: Change “for vertebrates” to “among vertebrate preys”

Reviewer 3 ·

Basic reporting

The work evaluated the diet, pellet parameters and availability of small mammals in one locality of Mediterranean Chile, during a long drought period. And the work compares their results with previous studies in the area.
The authors found diet parameters significant different than previous works, for example, Food Niche Breath values and high consumption of arachnids and they relate them with the extreme drought. However, in my opinion data that authors analyzed are not enough for this conclusion (30 pellets for each season for one year). I think that drought could be causing changed in the diet of B. magellanicus, but I consider the following points should be considered in this work for reaching this conclusion in a more robust way.
- The availability of arthropods was not assessed, so the authors do no know what is happening with arachnid communities and why B. magellanicus is consuming it in high frequency.
- It is not clear if drought is a natural event of it was caused by human impact.

Some other observations:
- Formulas and equations used for diet parameters and niche breath can be sent to supplementary material, as they are well known in the field of diet work.
- There are many cites in main text that are not included in the list of references

Experimental design

- For estimating small mammal abundance the authors trapped only two nights. At least three nights are needed for this kind of studies (You can see some examples: Heissler et al. 2016. Owl pellets: a more effective alternative to conventional trapping for broad-scale studies of small mammal communities. Methods in Ecology and Evolution 7: 96-103; Torres Contreras et al. 1994. Dieta y selectividad de presas de Speotyto cunicularia en una localidad semi- arid del norte de Chile a lo largo de siete años (1987-1993). Revista Chilena de Historia Natural 67: 329-340; Muñoz-Pedreros A., Fletcher A., Yáñez J. and Sánchez P.. 2010. Diversidad de micromamíferos en tres ambientes de la Reserva Nacional Lago Peñuelas, Región de - Valparaíso, Chile. Gayana 74(1): 1 – 11.
- Authors compare their finding with three works carried out in the area, but they omit the one of Jaksic F. and Marti C. 1984. Comparative Food Habits Of Bubo Owls In Mediterranean-Type Ecosystems. The Condor 86: 288-296.

Other comments were made on manuscript.

Validity of the findings

See point 1.

Additional comments

I annotated other comments on the manuscript

Annotated reviews are not available for download in order to protect the identity of reviewers who chose to remain anonymous.

---

## Round 0.2 · Minor Revisions

Kindly make the necessary changes as per the reviewer's suggestions.

·

Basic reporting

The manuscript is improved in according to the all reviewer comments.

Experimental design

It was explained with details and precise form the experimental desing.

Validity of the findings

The findings are valid, and interesting.

Additional comments

I suggest accept it.

Reviewer 2 ·

Basic reporting

In my opinion, issues have been addressed satisfactorily and the manuscript could be accepted after minor corrections:

Line 231: remove "for weight" and "for lengh"

Table 1: change the first "for" to "of" and the second "for" to "by".

Page 33: change Table 1 to Table 2. I believe that the number of tables were changed in the main text but not in attached tables, or something like that. Revise the number of each table in the PDF and look for the mistakes in the original files.

Title of the same table: change "for" to "of" (One-way ANOVA analysis of each morphological and weight variable of the pellets)

Table 3: change to "Similarity indices and unique fractions of hypervolumes by the different seasons..."

Table 4, second page: correct sp. to a roman font (not italics); and insert a foot with the meanings of the abbreviations (%F, %B)

Experimental design

it is ok.

Validity of the findings

they are ok.

Reviewer 3 ·

Basic reporting

I revised this manuscript by second time. Authors have attended most of suggestions and it was improved.

Experimental design

I still hold my opinion of first review

Validity of the findings

No comment

Additional comments

Line 74: This change is not reflected in manuscript
Lines 82, 86, 92 and 355: delete hyphen in Udrizar Sauthier surname and use space instead
Line 92: This change is not reflected in manuscript
Lines 187-191: check redaction, it is repetitive

---

## Round 0.3 · accepted · Accept

The manuscript is improved based on reviewer comments, my suggestion is to accept

Reviewer 2 ·

Basic reporting

It is ok

Experimental design

It is acceptable

Validity of the findings

They are valid